# Set Learning for Generative Information Extraction

**Jiangnan Li**[1,2], **Yice Zhang**[1,3], **Bin Liang**[1,2,4], **Kam-Fai Wong**[4], and **Ruifeng Xu**[1,2,3*]

[1] Harbin Insitute of Technology, Shenzhen, China
[2] Guangdong Provincial Key Laboratory of Novel Security Intelligence Technologies
[3] Peng Cheng Laboratory, Shenzhen, China
[4] The Chinese University of Hong Kong, Hong Kong, China
lijiangnan@stu.hit.edu.cn, zhangyc_hit@163.com, xuruifeng@hit.edu.cn

## Abstract

Recent efforts have endeavored to employ the sequence-to-sequence (Seq2Seq) model in Information Extraction (IE) due to its potential to tackle multiple IE tasks in a unified manner. Under this formalization, multiple structured objects are concatenated as the target sequence in a predefined order. However, structured objects, by their nature, constitute an unordered set. Consequently, this formalization introduces a potential order bias, which can impair model learning. Targeting this issue, this paper proposes a set learning approach that considers multiple permutations of structured objects to optimize set probability approximately. Notably, our approach does not require any modifications to model structures, making it easily integrated into existing generative IE frameworks. Experiments show that our method consistently improves existing frameworks on vast tasks and datasets.

## 1 Introduction

Information Extraction (IE) aims to identify structured objects from unstructured text (Paolini et al., 2021). Recently, many research efforts have been devoted to using sequence-to-sequence (Seq2Seq) models to solve IE tasks (Lu et al., 2021; Huguet Cabot and Navigli, 2021; Yan et al., 2021a,b; Josifoski et al., 2022). This generative approach enables the development of a universal IE architecture for different IE tasks (Lu et al., 2022; Fei et al., 2022; Lou et al., 2023). Moreover, combining the generative approach with Pre-trained Language Models (PLMs) has shown promising results in many IE tasks (Ma et al., 2022).

To formulate an IE task as a Seq2Seq problem, two steps are involved: (1) transforming each structured object into a flat sub-sequence, and (2) sorting and concatenating multiple sub-sequences according to a predefined order.

---

*  Corresponding Author

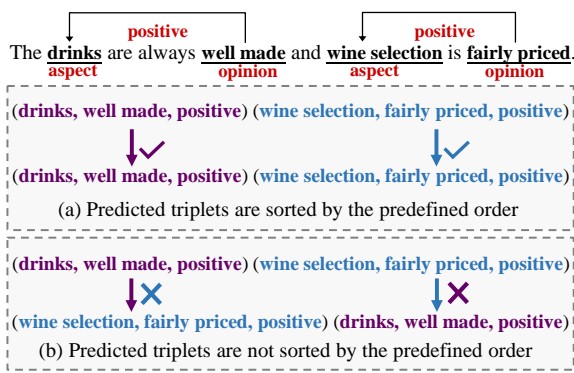

Figure 1: We use triplet extraction in aspect-based sentiment analysis as an example to illustrate the order bias. As depicted in (b), the model incurs a significant loss value despite correctly generating the triplets.

However, an essential aspect has been overlooked in these works, namely, that multiple structured objects constitute a set. As illustrated in Figure 1, assigning a predefined order introduces a harmful **order bias**, which violates the inherent unordered nature and makes the model lose its generalizability between permutations. Previous works have attempted to address this issue by modifying the decoder to generate multiple objects in an orderless and parallel manner (Sui et al., 2021; Tan et al., 2021; Ye et al., 2021; Mao et al., 2022), but these methods reduce model universality and make it difficult to combine with existing PLMs.

Our objective is to tackle order bias while preserving the advantages of generative IE. Inspired by Qin et al. (2019), we propose a novel approach called **set learning** for generative IE. The key idea of set learning is taking into account multiple possible orders to approximate the probability of structured objects set, thereby reducing order bias caused by only considering the predefined order.

Our approach is task-agnostic and does not necessitate any modification to model structures. These strengths enable the seamless integration of

set learning with existing off-the-shelf methods. We conduct extensive experiments in vast tasks and datasets. The results prove our approach can significantly and consistently improve current generative IE frameworks.

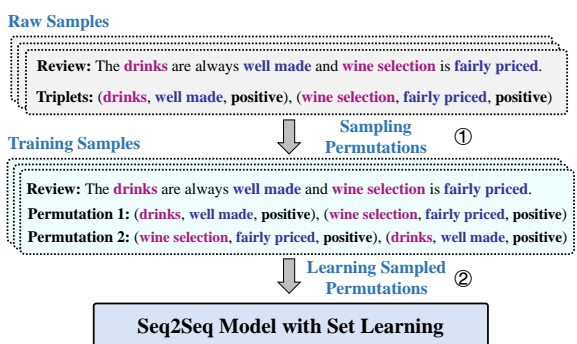

Figure 2: Overview of the proposed set learning approach. Our approach first samples permutations and then conducts set learning using sampled permutations.

## 2 Methodology

### 2.1 Generative IE and Seq2Seq learning

This section describes the general form of both IE tasks and generative IE.

Formally, an IE task generally takes a text $X = [x_1, x_2, \cdots]$ as input and outputs a set of structured objects $\mathbb{S} = \{s_1, s_2, \cdots\}$. Each structured object contains one or more spans from the input text, along with their types and relations.

**Generative IE** It usually takes two steps to transform an IE task into a generative paradigm: (1) flatten all elements of $\mathbb{S}$ into sub-sequences $\mathbb{Y}$, where $\mathbb{Y}$ is a set with the same size as $\mathbb{S}$; (2) concatenate sub-sequences of $\mathbb{Y}$ according to the predefined order $\pi^*$, resulting in permutation $\pi^*(\mathbb{Y})$. Here, $\pi^*(\mathbb{Y})$ is a flat sequence composed with tokens $[y_1, y_2, ...]$. With the above two steps, a raw sample is converted into a sequence pair $(X, \pi^*(\mathbb{Y}))$ that can be used to train Seq2Seq models.

Remarkably, most of the existing frameworks sort structured objects according to their positions within the input text since it generally achieves good performance (Yan et al., 2021a; Lu et al., 2022). Therefore, throughout the subsequent sections of this paper, we adopt the symbol $\pi^*$ to denote this sorting method and refer to it as "the reference permutation".

**Seq2Seq Learning** (Sutskever et al., 2014) is the default approach employed for training a genera-

tive IE framework. It decomposes the conditional probability $p(\pi^*(\mathbb{Y}) \mid X)$ using the chain rule:

$$\mathcal{L}_{\text{Seq2Seq}} = -\frac{1}{L} \log p(\pi^*(\mathbb{Y}) \mid X), \qquad (1)$$

$$= -\frac{1}{L} \sum_{t=1}^{L} \log p(y_t | Y_{<t}, X),$$

where $L = |\pi^*(\mathbb{Y})|$ denotes the number of target tokens, and $Y_{<t} = [y_1 y_2 \cdots y_{t-1}]$ denotes the sequence of target tokens up to the $t$-th position.

### 2.2 Proposed Set Learning Approach

The Seq2Seq learning paradigm optimizes generative models by maximizing $p(\pi^*(\mathbb{Y}) \mid X)$. However, both $\mathbb{S}$ and $\mathbb{Y}$ are inherently unordered sets. Consequently, solely optimizing for a single permutation introduces a bias, as indicated by the inequality expressed in Eq. 2:

$$p(\mathbb{S} \mid X) = p(\mathbb{Y} \mid X) \neq p(\pi^*(\mathbb{Y}) \mid X). \qquad (2)$$

To address this limitation, it is crucial to compute the set probability by considering all possible permutations, which can be formulated as follows:

$$p(\mathbb{S} \mid X) = p(\mathbb{Y} \mid X) = \sum_{\pi_z(\mathbb{Y}) \in \Pi(\mathbb{Y})} p(\pi_z(\mathbb{Y}) \mid X), \quad (3)$$

where $\Pi(\mathbb{Y})$ represents all permutations of $\mathbb{Y}$.

**Loss Function** Building upon Eq. 3, we define the following loss function. By minimizing this loss function, we can provide an unbiased optimization for the generative model.

$$\mathcal{L}_{\text{Set}} = -\log \left[ \sum_{\pi_z(\mathbb{Y}) \in \Pi(\mathbb{Y})} p(\pi_z(\mathbb{Y}) \mid X)^{\frac{1}{L}} \right], \quad (4)$$

where $\frac{1}{L}$ serves as a normalization term aimed at mitigating the impact of sequence length. In Appendix B, we analyze other available loss functions.

**Permutation Sampling** The factorial complexity of the size of $\Pi(\mathbb{Y})$ makes direct optimization of Eq. 4 computationally expensive. To overcome this challenge, we employ a sampling strategy to select a subset from $\Pi(\mathbb{Y})$. Specifically, we select the top-k permutations that are most similar to $\pi^*(\mathbb{Y})$ among all possible permutations. The similarity between permutations is computed using token-level edit distance. During model training, we substitute the full set of permutations in Eq. 4 with the sampled subset, effectively reducing the computational burden.

| Baseline | Unified ABSA | | | | | | | | REBEL | | | | Text2Event | | Unified NER | AVG △ |
|---|---|---|---|---|---|---|---|---|---|---|---|---|---|---|---|---|
| Backbone | BART-Base (Lewis et al., 2020) | | | | | | | | BART-Large | | | | T5-Base | | BART-Large | |
| Task | ABSA Triplet Extraction | | | | | | | | RE Triplet Extraction | | | | End-to-End EE | | NER | |
| Dataset | $D_{20a}$ | | | | $D_{20b}$ | | | | CoNLL | NYT | DocRED | ADE | ACE05 | | CADEC | |
| | 14res | 14lap | 15res | 16res | 14res | 14lap | 15res | 16res | | | | | T-C | A-C | | |
| Seq2Seq | 72.4 | 57.5 | 60.1 | 69.9 | 65.2 | 58.6 | 59.2 | 67.6 | 75.4 | 92.0 | 47.1 | 82.2 | 69.2 | 49.8 | 70.6 | - |
| Uniform | 73.0 | 57.8 | 62.6 | 72.5 | 71.1 | 58.0 | 61.9 | 69.1 | 74.2 | 91.7 | 45.9 | 81.9 | 68.0 | 47.9 | 70.0 | +0.59 |
| SetRNN | 71.1 | 56.6 | 59.2 | 68.9 | 63.2 | 57.8 | 59.2 | 67.1 | 76.6 | 92.2 | 47.8 | 82.4 | 68.7 | 48.7 | 69.7 | -0.51 |
| Set (Ours) | 73.4 | 60.8 | 63.5 | 74.4 | 71.7 | 58.7 | 62.2 | 70.6 | 76.8 | 92.2 | 48.2 | 82.9 | 69.6 | 51.5 | 72.2 | +2.10 |

| Baseline | UIE-SEL (T5-Large Backbone (Raffel et al., 2020)) | | | | | | | | AVG △ |
|---|---|---|---|---|---|---|---|---|---|
| Task | ABSA Triplet Extraction | | | | RE Triplet Extraction | | | | |
| Dataset | 14res | 14lap | 15res | 16res | CoNLL | NYT | SciERC | ACE05 | |
| Seq2Seq | 73.8 | 63.2 | 66.1 | 73.9 | 73.1 | 93.5 | 33.4 | 64.7 | - |
| Uniform | 73.9 | 63.0 | 66.2 | 73.4 | 73.4 | 93.2 | 32.7 | 64.1 | -0.23 |
| SetRNN | 73.5 | 62.6 | 65.3 | 73.1 | 73.0 | 92.8 | 33.1 | 63.7 | -0.58 |
| Set (Ours) | 74.9 | 63.5 | 67.5 | 74.7 | 73.7 | 93.5 | 35.9 | 65.9 | +0.99 |

Table 1: Experimental results. Our approach achieves improvements under different tasks and baselines.

## 3 Experiments

### 3.1 Experimental Setup

**Datasets** We conduct experiments on 9 widely used datasets across four 4 well-representative tasks of IE: Aspect-Based Sentiment Analysis (ABSA), Event Extraction (EE), Relation Extraction (RE), and Named Entity Recognition (NER). The used datasets include Semeval (Pontiki et al., 2014) $D_{20a}$ version (Peng et al., 2020) and Semeval $D_{20b}$ version (Xu et al., 2020) for ABSA; CoNLL04 (Roth and Yih, 2004), NYT (Riedel et al., 2010), ADE (Gurulingappa et al., 2012), DocRED (Yao et al., 2019), and SciERC (Luan et al., 2018) for RE; ACE2005 (Christopher et al., 2006) for EE and RE; CADEC (Karimi et al., 2015) for NER. We provide detailed statistics of these datasets in Appendix A.

**Baselines** We apply the proposed set learning approach to five representative frameworks. These frameworks include Unified ABSA (Yan et al., 2021b) for Aspect-Based Sentiment Analysis (ABSA), REBEL (Huguet Cabot and Navigli, 2021) for Relation Extraction (RE), Text2Event (Lu et al., 2021) for Event Extraction (EE), Unified NER (Yan et al., 2021b) for Named Entity Recognition (NER), and UIE-SEL (Lu et al., 2022) for ABSA and EE. For REBEL, we reproduce its pre-training using set learning. Since there is no available pre-training corpus for UIE-SEL, we compare its performance without pre-training.

**Implementation Details** To ensure a fair comparison, we introduce minimal modifications to the original frameworks when applying the proposed set learning approach to the baselines. Our modifications primarily focus on two aspects: (1) sampling permutations of the training data, and (2) adapting the existing Seq2Seq learning loss function of the frameworks to the proposed set learning loss function. Additionally, we utilize the micro $F_1$ score as the evaluation metric for assessing the performance of the models.

**Other Available Loss Functions** In addition to the proposed set learning loss function, we explore two alternative loss functions, which are originally designed for multi-label classification tasks.

(1) *SetRNN* (Qin et al., 2019) directly optimizes Eq. 3:

$$\mathcal{L}_{\text{SetRNN}} = -\log \left[ \sum_{\pi_z(\mathbb{Y}) \in \Pi(\mathbb{Y})} p(\pi_z(\mathbb{Y}) \mid X) \right]. \quad (5)$$

(2) *Uniform* (Vinyals et al., 2016) calculates the total negative log-likelihood of permutations:

$$\mathcal{L}_{\text{Uniform}} = -\sum_{\pi_z(\mathbb{Y}) \in \Pi(\mathbb{Y})} \frac{1}{L} \log p(\pi_z(\mathbb{Y}) \mid X). \quad (6)$$

### 3.2 Main Results

According to the result in Table 1, the proposed set learning approach consistently achieves improvements over five baselines across nine datasets, demonstrating its effectiveness and broad applicability. Moreover, the experimental findings reveal that the *Uniform* and *SetRNN* loss functions exhibit significant instability and, in many cases, lead to detrimental effects. In contrast, our proposed loss function consistently and significantly improves performance, highlighting its superiority over these alternative loss functions. A more detailed analysis and discussion of different loss functions can be found in Appendix B.

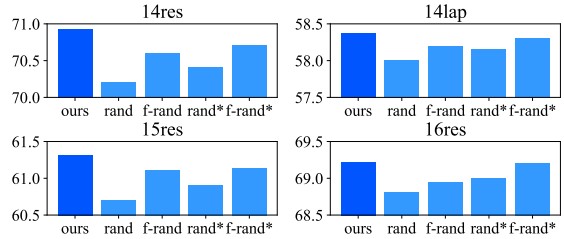

Figure 3: Comparisons of different sampling strategies. Vertical axes represent F1 scores.

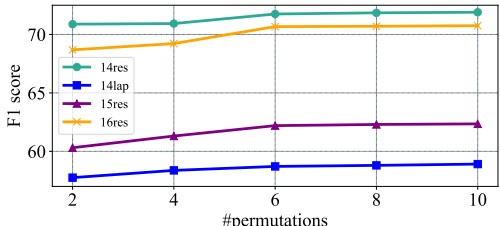

Figure 4: Effect of increasing permutations.

## 3.3 Further Analysis

**Sampling Strategy**    We analyze the effectiveness of the proposed sampling strategy by comparing it with four other strategies: random (rand), fixed-random (f-rand), random* (rand*), and fixed-random* (f-rand*). The random strategy randomly samples permutations for each epoch. The fixed-random strategy randomly samples a permutation for each training sample in the first epoch and fixes it for subsequent iterations. The random* and fixed-random* strategies are similar to random and fixed-random, respectively, but they always include the reference permutation.

As shown in Figure 3, the random strategy performs the worst, while the fixed-random strategy shows some improvement. Introducing the reference permutation enhances their performance, but they still lag significantly behind our sampling strategy. These results indicate the effectiveness of our sampling strategy.

**Number of Permutations**    We conduct experiments to investigate the impact of the number of permutations on the model performance. The results in Figure 4 indicate that the model performance improves as the number of permutations increases. This correlation implies that an increased number of permutations lead to a more accurate approximation of the set probability. Besides, we also find that once the number of permutations exceeds

**Case 2:**
**Review:** Our **waiter** was **helpful** and **charming**, the **food** was **perfect**, and the **wine** was **good**, too.
**Seq2Seq:** (food, perfect, POS) , (wine, good, POS),  (waiter, helpful, POS).
**Set Learning:** (waiter, charming, POS), (food, perfect, POS), (waiter, helpful, POS), (wine, good, POS).

Figure 5: Case study. The triplets in grey represent the correct triplets that Seq2Seq learning failed to generate. During the training phase, reference order sort objects according to their position in review (aspect term first).

| Method | 14res | 14lap | 15res | 16res |
|--------|-------|-------|-------|-------|
| Seq2Seq | 65.2 | 58.6 | 59.2 | 67.6 |
| Seq2Set | 64.5 | 55.2 | 57.8 | 64.3 |
| Seq2Set (RS) | 68.8 | 57.4 | 59.6 | 68.1 |
| **Set (Ours)** | **71.7** | **58.7** | **62.2** | **70.6** |

Table 2: Comparison with Seq2Set on Semeval $D_{20b}$ version. RS indicates reward shaping.

6, the incremental improvement in performance becomes slight, indicating that 6 is a good trade-off between computational cost and performance.

## 3.4 Case Study

We present illustrative examples in Figure 5 to facilitate a better understanding of the proposed approach. When confronted with examples that contain multiple structured objects, Seq2Seq learning may deviate from the reference permutation and omit some objects. This deviation occurs because decoding is based on likelihood rather than position. Thus solely learning a single permutation can easily result in sub-optimal predictions. In contrast, set learning has a stronger generalization across permutations, enabling the generation of objects in a more flexible order and recalling more objects.

## 3.5 Comparison with Order-Invariant Method

Another view for solving the order bias issue uses order-invariant loss, which is deeply explored in label generation tasks. A representative method is Seq2Set (Yang et al., 2019). To address the order bias in the label generation model, Seq2Set employs a reinforcement learning approach that utilizes the F1 score of the generated label sequence

as a reward for reinforcement fine-tuning. Since the F1 score is an order-invariant metric, Seq2Set reduces the impact of order bias in the label generation model.

We apply Seq2Set for generative information extraction to compare with our method. As illustrated in Table 2, the performance of Seq2Set is notably inferior. We speculate that the reason is that rewards in generative information extraction are more sparse and exhibit more significant variance. Furthermore, even after applying reward-shaping (Ng et al., 1999) techniques to enhance Seq2Set, the improvements in performance are still marginal compared to our approach.

Overall, our experiments show that order-invariant loss does not fit for generative information extraction, where task form is more complex than label generation tasks.

## 4 Related Work

Many studies have been conducted to explore how neural networks should be designed when dealing with sets as inputs or outputs. Vinyals et al. (2016) and Zaheer et al. (2017) proposed that neural networks should be permutations invariant to inputs when the inputs are sets. Vinyals et al. (2016) demonstrated that the permutation of outputs significantly influences the performance of RNN models when generating sets.

Tasks in NLP, such as multi-label classification and keyword generation, can be formulated as set generation tasks. Madaan et al. (2022) proposed a permutation sampling strategy for multi-label classification tasks, which uses the dependencies between labels to generate informative permutations. Mao et al. (2022) uses beam search to generate all triplets corresponding to an input simultaneously for sentiment triplet extraction tasks. Ye et al. (2021) proposed the One2Set model for keyword generation, which simultaneously generates all keywords corresponding to input through a parallel decoder.

In contrast, we propose a more sample and universal approach to optimize various existing generative IE frameworks. Our approach can easily combine with the off-the-shelf methods and achieve promising improvements.

## 5 Conclusion

In this paper, we reveal the order bias issue in generative Information Extraction (IE) and pro-pose the set learning approach for generative IE to address this issue. The proposed set learning approach defines and optimizes set probability in Seq2Seq models and reduces the computational cost by permutation sampling. Notably, the proposed approach can be easily integrated with existing generative IE frameworks as a plugin to enhance performance. Experiments demonstrate that the proposed approach significantly improves existing frameworks in various tasks and datasets. We believe our work can inspire further research on the potential of set learning in other natural language processing tasks.

## Limitations

One important limitation of our approach is that users need to perform a trade-off between performance and computational time consumption. When the size of training data is relatively small, the time consumption will concentrate on the inference phase; an increase in the time consumption of the training phase is negligible compared to a notable increase in performance. However, when the training data is relatively large, sampling a large number of permutations for each sample may only result in a marginal improvement but will significantly lengthen training time.

Additionally, an important research challenge lies in accurately estimating set probability during the inference phase of Seq2Seq models. Seq2Seq models use greedy decoding or beam search for inference, which is based on sequence probabilities rather than set probabilities. However, the sequence with maximum probability does not necessarily correspond to the set with maximum probability (Qin et al., 2019). Therefore, investigating novel approaches that provide more accurate estimations of set probability is a valuable direction for future research.

## Acknowledgements

We thank the anonymous reviewers for their valuable suggestions to improve the quality of this work. This work was partially supported by the National Natural Science Foundation of China (62006062, 62176076), Natural Science Foundation of GuangDong 2023A1515012922, Shenzhen Foundational Research Funding JCYJ20220818102415032, Guangdong Provincial Key Laboratory of Novel Security Intelligence Technologies 2022B1212010005.

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

## A Dataset Statistics and Reproducibility

The hyperparameters used in all experiments are provided in Table 3. For the experiments conducted on the UIE-SEL framework (Lu et al., 2022), we were constrained by computational resources and did not perform hyperparameter tuning. Instead, we used the original hyperparameters and set the number of permutations to 2.

Detailed statistics of the datasets used in our experiments are presented in Table 4. We have ensured that our use of the datasets falls within their intended scope and aligns with existing works.

| Dataset | learning_rate | adam_epsilon | batch_size | accumulated_steps | max_epochs | seeds | lr_scheduler | #sampled permutations |
|---------|---------------|--------------|------------|-------------------|------------|-------|--------------|-----------------------|
| $\mathcal{D}_{20a}$ | 1e-4 | 1e-8 | 16 | 1 | 100 | 0, 1, 2, 3 | linear | 6 |
| $\mathcal{D}_{20b}$ | 1e-4 | 1e-8 | 16 | 1 | 100 | 0, 1, 2, 3 | linear | 6 |
| CADEC | 2e-5 | 1e-8 | 8 | 1 | 30 | 0, 1, 2, 3 | linear | 4 |
| ACE 2005 EE | 1e-4 | 1e-8 | 16 | 1 | 30 | 41, 42, 43, 44 | linear | 4 |
| ConLL04 | 1e-4 | 1e-8 | 8 | 4 | 35 | 41, 42, 43, 44 | linear | 4 |
| NYT | 5e-5 | 1e-8 | 8 | 3 | 40 | 41, 42, 43, 44 | linear | 2 |
| DocRED | 2e-5 | 1e-8 | 2 | 16 | 20 | 41, 42, 43, 44 | linear | 4 |
| ADE | 5e-5 | 1e-8 | 8 | 4 | 25 | 42 (10 fold) | linear | 4 |

Table 3: Hyperparameters used for all the experiments.

## B  Discussion of Loss Functions

In this section, our primary focus is to analyze the probability distributions generated by different loss functions among permutations. This analysis is essential for comprehending the performance variations observed with these loss functions.

**Uniform Loss**  A possible solution for optimising Eq. 3 is to optimize all permutations equally (Vinyals et al., 2016). This can be achieved by *Uniform* loss function:

$$\mathcal{L}_{\text{Uniform}} = -\sum_{\pi_z(\mathbb{Y}) \in \Pi(\mathbb{Y})} \frac{1}{L} \log p(\pi_z(\mathbb{Y}) \mid X). \quad (7)$$

Since the *Uniform* loss calculates the negative log-likelihood of each permutation, it imposes a significant penalty on permutations with low probability. Consequently, ***Uniform* results in a uniform distribution over permutations**. It can also be demonstrated that the minimum value of *Uniform* is attained when the probabilities of all permutations are equal.

When optimizing *Uniform*, a crucial insight is that the total probability of $\mathbb{Y}$ across all permutations should be less than or equal to 1. Thus, our optimization problem aims to minimize Eq. 7 while satisfying the following constraint:

$$\sum_{\pi_z(\mathbb{Y}) \in \Pi(\mathbb{Y})} p(\pi_z(\mathbb{Y}) \mid X) = 1. \quad (8)$$

Under the constraint mentioned above, we can formulate the optimization of *Uniform* as an unconstrained problem using the Lagrange multiplier method in the following manner:

$$L = -\sum_{\pi_z(\mathbb{Y}) \in \Pi(\mathbb{Y})} \frac{1}{|\pi_z(\mathbb{Y})|} \log p(\pi_z(\mathbb{Y}) \mid X)$$
$$+ \lambda \left[ \sum_{\pi_z(\mathbb{Y}) \in \Pi(\mathbb{Y})} p(\pi_z(\mathbb{Y}) \mid X) - 1 \right]. \quad (9)$$

where $\lambda$ is a Lagrange multiplier, and $L$ is a function that depends on both $\lambda$ and each $p(\pi_z(\mathbb{Y}))$ for

$\pi_z(\mathbb{Y}) \in \Pi(\mathbb{Y})$. Solving Equation 9 involves finding the minimum point, which requires that each $\pi_z(\mathbb{Y}) \in \Pi(\mathbb{Y})$ has the same probability.

However, in Seq2Seq models, the probability of permutations can be influenced by various factors (Vinyals et al., 2016; Madaan et al., 2022), leading to an inherent imbalance in the distribution of permutations. This indicates that fitting a uniform distribution may be challenging. Therefore, imposing a strict penalty on non-uniformity, such as the **rigorous penalty of *Uniform***, may result in subpar performance (Qin et al., 2019).

**SetRNN Loss**  The *SetRNN* loss, proposed by Qin et al. (2019), offers an alternative approach by directly optimizing the set probability (Eq. 3). Instead of computing the negative log-likelihood of each permutation, *SetRNN* loss calculates the total probability of all permutations and utilizes it to compute the loss:

$$\mathcal{L}_{\text{SetRNN}} = -\log \left[ \sum_{\pi_z(\mathbb{Y}) \in \Pi(\mathbb{Y})} p(\pi_z(\mathbb{Y}) \mid X) \right]. \quad (10)$$

Unlike the *Uniform* loss, the *SetRNN* loss does not assume uniformity among permutations since the distribution is not explicitly considered in the total probability.

However, a drawback of the *SetRNN* loss is that it does not penalize non-uniformity, which can lead the model to take shortcuts and assign almost all of the probability density to a single permutation. Our experiments have shown that such shortcuts are prevalent when using the *SetRNN* loss, which may explain its poor performance compared to other loss functions..

**Set Loss**  We propose the *Set* loss, which is derived from the *Uniform* loss but relaxes the penalty for low-probability permutations to support set learning better.

We start by transforming the *Uniform* loss into the following form:

$$\mathcal{L}_{\text{Uniform}} = -\sum_{\pi_z(\mathbb{Y})\in\Pi(\mathbb{Y})} \log p(\pi_z(\mathbb{Y}) \mid X)^{\frac{1}{L}}. \quad (11)$$

By moving the $\log$ outside the summation, we can relax the penalty and obtain the *Set* loss:

$$\mathcal{L}_{\text{Set}} = -\log\left[\sum_{\pi_z(\mathbb{Y})\in\Pi(\mathbb{Y})} p(\pi_z(\mathbb{Y}) \mid X)^{\frac{1}{L}}\right], \quad (12)$$

The *Set* loss reduces the penalty while maintaining a looser uniformity restriction. Therefore, it does not suffer from the "rigorous penalty" or "taking shortcuts" issues observed in other loss functions. The probability distribution produced by the *Set* loss falls in a middle state between the probability distributions produced by the *Uniform* and *SetRNN* losses.

**Validation Experiment** We design and implement an experiment to corroborate the arguments in the above discussions. Specifically, we use the $D_{20b}$ version of the 14lap dataset and set the number of permutations to 2, optimize the model to convergence using different losses, and then record and analyze the probability distributions generated by the different losses. Our main findings are as follows: (1) *SetRNN* loss assigns more than 0.7 probability mass to a single permutation with a 97% probability, while the other two losses have a probability of 0%. (2) *Uniform* loss assigns a probability mass of 0.48 to 0.52 to both permutations with a probability of 26%. (3) *Set* loss assigns a probability mass of 0.48 to 0.52 to both permutations with a probability of 21%.

The results are consistent with our claims: *SetRNN* loss assigns much more probability density to a single permutation, and *Uniform* loss produces a more uniform distribution. In contrast, *Set* loss relaxes the uniformity of *Uniform* loss.

| Dataset | #Samples | #Structured Objects | #Object Types |
|---|---|---|---|
| $\mathcal{D}_{20a}$ 14res train | 1,300 | 2,145 | 3 |
| $\mathcal{D}_{20a}$ 14res dev | 323 | 524 | 3 |
| $\mathcal{D}_{20a}$ 14res test | 496 | 862 | 3 |
| $\mathcal{D}_{20a}$ 14lap train | 920 | 1,265 | 3 |
| $\mathcal{D}_{20a}$ 14lap dev | 228 | 337 | 3 |
| $\mathcal{D}_{20a}$ 14lap test | 339 | 490 | 3 |
| $\mathcal{D}_{20a}$ 15res train | 593 | 923 | 3 |
| $\mathcal{D}_{20a}$ 15res dev | 148 | 238 | 3 |
| $\mathcal{D}_{20a}$ 15res test | 318 | 455 | 3 |
| $\mathcal{D}_{20a}$ 16res train | 842 | 1,289 | 3 |
| $\mathcal{D}_{20a}$ 16res dev | 210 | 316 | 3 |
| $\mathcal{D}_{20a}$ 16res test | 320 | 465 | 3 |
| $\mathcal{D}_{20b}$ 14res train | 1,266 | 2,338 | 3 |
| $\mathcal{D}_{20b}$ 14res dev | 310 | 577 | 3 |
| $\mathcal{D}_{20b}$ 14res test | 492 | 994 | 3 |
| $\mathcal{D}_{20b}$ 14lap train | 906 | 1,460 | 3 |
| $\mathcal{D}_{20b}$ 14lap dev | 219 | 346 | 3 |
| $\mathcal{D}_{20b}$ 14lap test | 328 | 543 | 3 |
| $\mathcal{D}_{20b}$ 15res train | 605 | 1,013 | 3 |
| $\mathcal{D}_{20b}$ 15res dev | 148 | 249 | 3 |
| $\mathcal{D}_{20b}$ 15res test | 322 | 485 | 3 |
| $\mathcal{D}_{20b}$ 16res train | 857 | 1,394 | 3 |
| $\mathcal{D}_{20b}$ 16res dev | 210 | 339 | 3 |
| $\mathcal{D}_{20b}$ 16res test | 326 | 514 | 3 |
| ACE 2005 train (EE) | 17,172 | 4,202 | 33 |
| ACE 2005 dev (EE) | 923 | 450 | 33 |
| ACE 2005 test (EE) | 832 | 403 | 33 |
| ACE 2005 train (RE) | 10,051 | 4,788 | 6 |
| ACE 2005 dev (RE) | 2,420 | 1,131 | 6 |
| ACE 2005 test (RE) | 2,050 | 1,151 | 6 |
| CADEC | 7,597 | 6,318 | 1 |
| CoNLL04 train | 922 | 1,290 | 5 |
| CoNLL04 dev | 231 | 343 | 5 |
| CoNLL04 test | 288 | 422 | 5 |
| NYT train | 56,196 | 94,222 | 24 |
| NYT dev | 5,000 | 8,489 | 24 |
| NYT test | 5,000 | 8,616 | 24 |
| DocRED train | 3,008 | 37,486 | 96 |
| DocRED dev | 300 | 3,678 | 96 |
| DocRED test | 700 | 8,787 | 96 |
| SciERC train | 1,861 | 3,219 | 7 |
| SciERC dev | 275 | 455 | 7 |
| SciERC test | 551 | 974 | 7 |
| ADE (10 fold) | 4,272 | 6,821 | 1 |

Table 4: Datasets statistics.