# OpenReview forum: "Set Learning for Generative Information Extraction"
_EMNLP/2023/Conference — EMNLP 2023 Main_

### Official Review · Reviewer_wMdM · 2023-08-01

**Soundness:** 3

**Excitement:**

4: Strong: This paper deepens the understanding of some phenomenon or lowers the barriers to an existing research direction.

**Paper Topic And Main Contributions:**

This paper tackles the problem of order bias in Information Extraction (IE) frameworks using sequence-to-sequence (Seq2Seq) models. Order bias arises due to the processing of structured objects in a predefined order. To address this, the paper introduces a set learning approach that considers multiple permutations of structured objects to approximate the set probability, reducing order bias. The main contributions are:
1. Highlighting the issue of order bias in Seq2Seq-based generative IE.
2. Proposing a novel set learning approach that can be integrated into existing frameworks without modifications.
3. Validating the approach with experiments, showing consistent improvements across various IE tasks and datasets.

**Questions For The Authors:**

Please see my comment in the previous "Reasons to Reject" section.

**Reasons To Accept:**

1. Novel Solution: The paper proposes a unique 'set learning' solution to the problem. This solution, which incorporates multiple permutations of structured objects, innovatively addresses the problem without necessitating changes to existing model structures.
2. Experimental Validation: The effectiveness of the proposed approach is empirically validated using various tasks and datasets, showcasing its broad applicability and robustness.
3. Framework for Further Research: The proposed set learning approach could inspire new research not only in IE but also in other NLP tasks where order bias might be a concern. The paper could also serve as a basis for further explorations into other potential improvements in set learning and its applications in NLP.

**Reasons To Reject:**

1. Lack of Comparative Analysis: In the experiments, the paper does not include a comparison with other existing methods for dealing with order bias in IE tasks, but only compares with the baseline seq2seq approach and some variations to the proposed method. A more thorough comparison could provide a stronger argument for the effectiveness of their approach. If the results are not comparable, the author should elaborate on the reasons.
2. Shifting parts of Appendix B into the main paper would add depth to the discussion, particularly on why the proposed loss function outperforms the variations. Adding experimental results to validate the claims about the Uniform loss, SetRNN loss, and the proposed Set loss would also strengthen the paper. Specifically, for the Uniform loss, an experiment demonstrating that it indeed results in a uniform distribution over permutations would validate the authors' claim and offer a concrete understanding of the Uniform loss's behavior. Similarly, for SetRNN loss, showing experimentally that the model tends to assign most of the probability density to a single permutation would provide clear evidence supporting their discussion.

**Reproducibility:**

5: Could easily reproduce the results.

**Reviewer Confidence:**

4: Quite sure. I tried to check the important points carefully. It's unlikely, though conceivable, that I missed something that should affect my ratings.

---

> ### Author Rebuttal · Authors · 2023-08-28
>
> Thank you for your constructive feedback and suggestions. They are instrumental in enhancing our paper.
>
> **Weakness-1: Lack of some comparative analysis.**
>
> Response-1: Thank you kindly for bringing this matter to our attention. We agree that a more thorough comparison could provide a stronger argument for the effectiveness of our work. We will refine our paper by adding comparisons and analysis.
>
> Presently, there are two prevalent strategies to tackle order bias in generative methods:
>
> 1.	The first is modifying the model structure to generate multiple results in parallel, represented by Sui, Dianbo, et al. [1] and Ye, Jiacheng, et al. [2].
>
> 2.	The other one is modifying the training objective of the generative model, represented by Seq2Set [3].
>
> The first approach is difficult to integrate directly with existing generative information extraction methods. We add the second method for comparison.
>
> The table below shows that Seq2Set is underperformed compared to the base Seq2Seq model. The sparse and high variance rewards in generative information extraction tasks may cause this. Even though we improve it with reward shaping, it performs much worse than our method.
>
> | Method                    | 14res  | 14lap | 15res | 16res |
> | --------------------------| ------ | ----- | ----- | ----- |
> | Seq2Seq                   | 65.2   | 58.6  | 59.2  | 67.6  |
> | Seq2Set [1]               | 64.5   | 55.2  | 57.8  | 64.3  |
> | Seq2Set (reward-shaping)  | 68.8   | 57.4  | 59.6  | 68.1  |
> | **Set (Ours)**         | 71.7   | 58.7  | 62.2  | 70.6  |
>
> **Weakness-2: Appendix B should be supplemented with experiments and shifted to the main paper.**
>
> Response-2: We are very grateful for such insightful suggestions. We believe this will deepen the discussion and make our paper more engaging. We will move Appendix B to the main paper and add experiments to validate the theoretical analysis.
>
> In addition, we design and implement experiments to corroborate the arguments in Appendix B. Specifically, we set the number of permutations to 2, optimize the model to convergence using different losses, and then record and analyze the probability distributions generated by the different losses. Our main findings are as follows:
>
> 1.	SetRNN loss assigns more than 0.7 probability mass to a single permutation with a 97% probability, while the other two losses have a probability of 0%.
>
> 2.	Uniform loss assigns a probability mass of 0.48-0.52 to both permutations with a probability of 26%.
>
> 3.	Set loss assigns a probability mass of 0.48-0.52 to both permutations with a probability of 21%.
>
> The results are consistent with our claims in Appendix B: SetRNN loss assigns much more probability density to a single permutation, and Uniform loss produces a more uniform distribution, while Set loss relaxes the uniformity of Uniform loss. We will incorporate the complete experiment into our refined paper.
>
> References:
>
> [1] Sui, Dianbo, et al. "Set generation networks for end-to-end knowledge base population." Proceedings of the 2021 Conference on Empirical Methods in Natural Language Processing. 2021.
>
> [2] Ye, Jiacheng, et al. "One2Set: Generating Diverse Keyphrases as a Set." Proceedings of the 59th Annual Meeting of the Association for Computational Linguistics and the 11th International Joint Conference on Natural Language Processing (Volume 1: Long Papers). 2021.
>
> [3] Yang, Pengcheng, et al. "A deep reinforced sequence-to-set model for multi-label classification." Proceedings of the 57th Annual Meeting of the Association for Computational Linguistics. 2019.

---

### Official Review · Reviewer_rpsn · 2023-08-04

**Typos Grammar Style And Presentation Improvements:** 1. "four 4" --> "4" in line 140
2. Fi…
**Soundness:** 4

**Excitement:**

4: Strong: This paper deepens the understanding of some phenomenon or lowers the barriers to an existing research direction.

**Missing References:**

N/A

**Paper Topic And Main Contributions:**

This paper studies the order bias problem of the sequence-to-sequence (Seq2Seq) model in Information Extraction (IE). The authors propose a novel set learning method to address this potential bias. It considers multiple possible orders to approximate the probability of structured objects set. Specifically, the authors design a loss function and propose a customized sampling strategy for reducing computational costs.  Experiments show the effectiveness of the proposed method.



**Reasons To Accept:**

1. Well-motivated and clear writing
2. A novel method that considers order bias for the generative IE. I like the way that the authors highlight the issue
3. Extensive experiments on 9 datasets across 4 tasks.

**Reasons To Reject:**

1. the computations of permutation sampling may be expensive for long text, which involves more entities and relations.
2. the example in the introduction does not clearly demonstrate the order bias issue. I suggest the authors select another example for such an illustration.


**Reproducibility:**

4: Could mostly reproduce the results, but there may be some variation because of sample variance or minor variations in their interpretation of the protocol or method.

**Reviewer Confidence:**

4: Quite sure. I tried to check the important points carefully. It's unlikely, though conceivable, that I missed something that should affect my ratings.

---

> ### Author Rebuttal · Authors · 2023-08-28
>
> We greatly appreciate your insightful feedback and recommendations, which have been invaluable in refining our paper.
>
> **Weakness-1: The computations may be expensive for long text.**
>
> Response-1: Thank you for bringing this up. Indeed, our method does lead to an increase in computational cost, which is an inherent limitation of our approach. We hope further work can address this limitation.
>
> For texts with a larger number of tuples, while using more permutations in principle can better model the probability of the set, sampling a limited number of permutations can still offer significant benefits. For instance, with the DocRED dataset, where an individual sample contains an average of 12 tuples, we observed a 1.1 F1 score improvement even when sampling only four permutations.
>
> **Weakness-2: The example in the introduction is not clear.**
>
> Response-2: Thank you for highlighting this; it is instrumental in enhancing the presentation quality of our paper. We will replace it with an appropriate example to illustrate the order bias issue.
>
> **Typos Grammar Style And Presentation Improvements.**
>
> Thank you for pointing out these issues. We will make corrections and conduct further checks to minimize such errors.

---

### Official Review · Reviewer_sCUS · 2023-08-05

**Soundness:** 3

**Excitement:**

3: Ambivalent: It has merits (e.g., it reports state-of-the-art results, the idea is nice), but there are key weaknesses (e.g., it describes incremental work), and it can significantly benefit from another round of revision. However, I won't object to accepting it if my co-reviewers champion it.

**Paper Topic And Main Contributions:**

To tackle the potential order bias resulting from a predefined fixed target order in the previous sequence-to-sequence model of information extraction, this paper delves into the set learning approach. It introduces a novel loss function tailored for the generative information extraction task. Notably, their method is task-agnostic, and they conduct extensive experiments across various information extraction tasks and datasets.

**Questions For The Authors:**

Why do Uniform and SetRNN losses perform worse than the base seq2seq model?
Did you do significant tests or average multiple runs?

**Reasons To Accept:**

1. The method description is clear， and the experiments are sufficient.
2. Their analysis of the proposed method is detailed, and they experiment with various loss functions and permutation sampling.

**Reasons To Reject:**

1. I think innovativeness is somewhat lacking in their approach. Their method only attempts a slight refinement of the previously proposed set-learning loss function.

2. The order bias of the generative model has been explored in previous work using reinforcement learning in other tasks, I think these work should be analyzed and compared.

3. I think it is interesting to explore the problem of order bias when generative targets are “sets”. So it will be more convincing if the paper can find more advanced strategies and be extended to a long paper.

**Reproducibility:**

4: Could mostly reproduce the results, but there may be some variation because of sample variance or minor variations in their interpretation of the protocol or method.

**Reviewer Confidence:**

4: Quite sure. I tried to check the important points carefully. It's unlikely, though conceivable, that I missed something that should affect my ratings.

---

> ### Author Rebuttal · Authors · 2023-08-28
>
> We extend our gratitude for your observations and comments, which have been influential in enhancing the quality of our paper.
>
> **Weakness-1: The innovativeness is somewhat lacking.**
>
> Response-1: We sincerely appreciate your constructive comments concerning our work's perceived lack of innovation. We are grateful for this opportunity to underscore its merits.
>
> We would like to highlight that our core contribution is developing a simple but effective approach to mitigating order bias in existing generative information extraction methods. While this approach may seem simplistic at first glance, its true value lies in its ability to integrate with current models easily. This feature alone enhances its innovative appeal and has been acknowledged by two other reviewers.
>
> Moreover, our approach is not limited to the specific task at hand; it also shows promise for adaptation in a wide range of applications where order bias presents challenges or where task objectives involve unordered sets. Such a broad applicability amplifies the impact and relevance of our contributions.
>
> We hope this clarifies the innovative aspects of our work.
>
> **Weakness-2: The paper lacks an analysis and comparison of reinforcement learning methods.**
>
> Response-2: We are grateful for your insightful observation and constructive suggestions. We agree that incorporating an analysis and comparing reinforcement learning methods would significantly enhance our work. Consequently, we are committed to enriching our manuscript by adding this analysis and comparison. Below, we outline some initial experimental data along with corresponding analytical insights.
>
> | Method                    | 14res  | 14lap | 15res | 16res |
> | --------------------------| ------ | ----- | ----- | ----- |
> | Seq2Seq                   | 65.2   | 58.6  | 59.2  | 67.6  |
> | Seq2Set [1]               | 64.5   | 55.2  | 57.8  | 64.3  |
> | Seq2Set (reward-shaping)  | 68.8   | 57.4  | 59.6  | 68.1  |
> | **Set (Ours)**         | 71.7   | 58.7  | 62.2  | 70.6  |
>
> Seq2Set [1] uses order-invariant reinforcement learning to mitigate the order bias issue in multi-label generation tasks. However, when applied to generative information extraction, our results indicate that Seq2Set underperforms. We speculate that rewards in generative information extraction are more sparse and have a larger variance. Moreover, even after leveraging reward-shaping techniques to improve Seq2Set, the gains in performance remain marginal compared to our approach.
>
>
>
> **Q1: What leads to the Uniform and SetRNN losses underperforming?**
>
> A1: We are glad you highlighted this. A detailed analysis can be found in Appendix B, which we will supplement with additional experimental data. In essence, both Uniform loss and SetRNN loss tend to generate pathological probability distributions over the chosen permutations, which contradicts the intrinsic nature of generative information extraction tasks and set learning.
>
> Specifically, due to the lack of regularization on probability distributions over permutations, SetRNN loss causes the model to take shortcuts to assign most of the probabilities to single permutations, which is contrary to the essence of set learning: learning multiple permutations to make the model generalize between permutations. Conversely, Uniform loss over-penalizes any non-uniform distribution, which is over-strict for the Seq2Seq model, which may have inherent order bias. These contradictions might primarily account for their poor results.
>
> **Q2: Are there significant tests and multiple run averages?**
>
> A2: All of our experiments were repeated four times, and average results were reported. In addition, we test the significance of our method against the base Seq2Seq model and the variants of our method, and the results show that our method significantly outperformed the others (p-value < 0.0001). We will integrate this information into our revised manuscript for better comprehensibility.
>
> References:
>
> [1] Yang, Pengcheng, et al. "A deep reinforced sequence-to-set model for multi-label classification." Proceedings of the 57th Annual Meeting of the Association for Computational Linguistics. 2019.

---

### Meta-Review · Area_Chair_vKPb · 2023-09-24

**Recommendation:** 4

**Metareview:**

The paper introduces a novel 'set learning' approach to address order bias in Seq2Seq models for IE. Experiments across various IE tasks and datasets demonstrates the effectiveness and applicability of the proposed approach. The paper has the potential to inspire further research in IE and related NLP tasks. However, a more comprehensive comparative analysis with existing order bias mitigation methods is needed to strengthen the paper's claims. Additional experimental evidence is recommended to validate claims related to the proposed Set loss and various other losses. Overall, the paper shows promise but should address these areas in the next version.

---

### Decision · Program_Chairs · 2023-10-07

**Decision:**

Accept-Main

**Comment:**

The paper introduces a novel 'set learning' approach to address order bias in Seq2Seq models for IE. Experiments across various IE tasks and datasets demonstrates the effectiveness and applicability of the proposed approach. The paper has the potential to inspire further research in IE and related NLP tasks. However, a more comprehensive comparative analysis with existing order bias mitigation methods is needed to strengthen the paper's claims. Additional experimental evidence is recommended to validate claims related to the proposed Set loss and various other losses. Overall, the paper shows promise but should address these areas in the next version.